# Herbal Cannabis and Depression: A Review of Findings Published over the Last Three Years

**DOI:** 10.3390/ph17060689

**Published:** 2024-05-27

**Authors:** Jozsef Haller

**Affiliations:** 1Drug Research Institute, 1137 Budapest, Hungary; haller.jozsef@uni-nke.hu; 2Department of Criminal Psychology, Faculty of Law Enforcement, Ludovika University of Public Service, 1083 Budapest, Hungary

**Keywords:** herbal cannabis, depression, pharmacological plausibility

## Abstract

Public perception contrasts scientific findings on the depression-related effects of cannabis. However, earlier studies were performed when cannabis was predominantly illegal, its production was mostly uncontrolled, and the idea of medical cannabis was incipient only. We hypothesized that recent changes in attitudes and legislations may have favorably affected research. In addition, publication bias against cannabis may have also decreased. To investigate this hypothesis, we conducted a review of research studies published over the last three years. We found 156 relevant research articles. In most cross-sectional studies, depression was higher in those who consumed cannabis than in those who did not. An increase in cannabis consumption was typically followed by an increase in depression, whereas withdrawal from cannabis ameliorated depression in most cases. Although medical cannabis reduced depression in most studies, none of these were placebo-controlled. In clinical studies published in the same period, the placebo also ameliorated depression and, in addition, the average effect size of the placebo was larger than the average effect size of medical cannabis. We also investigated the plausibility of the antidepressant effects of cannabis by reviewing molecular and pharmacological studies. Taken together, the reviewed findings do not support the antidepressant effects of herbal cannabis.

## 1. A Brief Historical Overview

Medical uses of cannabis are ancient; the first Chinese and Indian accounts of its hypnotic, pain-relieving and anxiolytic properties date back to the first century BC [1]. In Europe and North America, the plant was used mainly to produce fibers till the 19th century, when its psychoactive properties were rediscovered [2]. Beyond becoming a widespread recreational drug, cannabis was also used to treat insomnia, pain, and asthma. In addition, it was consumed to treat several other conditions and diseases, including depression. Among the patients, we find notable personalities such as Victoria, Queen of England, and Empress Elizabeth of Austria-Hungary [3]. This first golden age of medicinal cannabis use, however, came to an end at the beginning of the 20th century, when new legislations made its use more and more difficult. The first trial to limit cannabis was the Marihuana Tax Act in 1937 [4]. This was primarily motivated by societal worries, but medical and criminological concerns were also considered. The process culminated in the 1961 United Nations Single Convention on Narcotic Drugs that placed cannabis under strict control, with the aim of protecting the physical and psychological health of the population [5]. Concerns related to the addictive properties of herbal cannabis were reiterated in this convention as well.

The discovery of the active constituents of herbal cannabis, and their mechanisms of action in late 20th–early 21st century (see Section 4 for details) gave a great impetus to cannabinoid research. It was soon suggested that the endocannabinoid system may become an important target of drug development for a series of medical conditions, among others, for the development of a novel class of antidepressants [6]. In parallel with scientific development, a second golden age of medical cannabis use emerged, gradually leading to its widespread liberalization. The basic idea of this movement was that natural cannabis is per se appropriate to treat a variety of diseases, and patients should be allowed to self-medicate themselves by lifting the bans on cannabis.

In general terms, the main motives of cannabis consumption are the enhancement of positive emotions, the improvement of social events or relationships, the promotion of coping with anxiety and depression, and the enhancement of experience and creativity [7]. As such, two of the main motives are related to the improvement of mental health, which potentially makes the argumentation of the UN Single Convention on Narcotic Drugs obsolete. Depression—the focus of this review—is a major public health concern, which constitutes a high economic burden to the society, dramatically increases the risk of suicide, and markedly impairs quality of life [8]. Despite great advancements in the treatment of depression, antidepressant medications still suffer from shortcomings [9]. They are not always effective, have severe side effects in many patients, and may interact with other medications administered for comorbid diseases. As such, any novel treatment opportunity for depression is highly valuable from the point of view of public health.

There is an abundance of material on the Internet that claims that cannabis effectively treats depression, from professionally conducted exploratory research [10], through blogs and Internet archives summarizing the personal experiences of clinicians and general practitioners [11,12,13], to the dedicated writings of activists fighting for cannabis legalization [14]. Their point of view was most succinctly formulated as follows: “The power of cannabis to fight depression is perhaps its most important property” ([15], p. 58).

Despite this enthusiasm, however, the antidepressant effects of cannabis are controversial at best. Reviews covering findings obtained over the last two to three decades conclude in a reasonably consistent fashion that herbal cannabis aggravates rather than ameliorates mood disorders and depression in particular [16,17,18,19,20,21]. In addition, early cannabis use may lower the age of onset of mood disorders and increase suicidality [22,23,24,25].

We hypothesized, however, that recent debates over cannabis increased public awareness, the legalization of cannabis promoted the use of licensed products and decreased publication bias against cannabis, and, in addition, the medically supervised use of cannabis improved consumption habits. All these developments may have favorably affected the use of cannabis to ameliorate depression. To investigate this hypothesis, we reviewed research studies published over the last three years.

## 2. The Selection of Studies for Review

We searched the PubMed database by using the following search term: (marihuana [title/abstract] OR marijuana [title/abstract] OR cannabis [title/abstract]) AND (depression [title/abstract] OR depressive [title/abstract] OR bipolar [title/abstract] OR mood [title/abstract]). The search returned 1143 hits for the last three years, e.g., for the period between February 2021 and February 2024. We selected for the review studies that met the following criteria: published in English; research undertaken in humans; reported research findings together with clearly described methodologies; depression studied by a validated methodology, e.g., by psychometric instruments or by diagnostic criteria defined by the Diagnostic and Statistical Manual of Mental Disorder or the International Classification of Diseases; study participants consumed herbal cannabis; the relationship between depression symptoms and cannabis consumption was studied. We excluded studies if the consequences of cannabis consumption could not be differentiated from those of other drugs (e.g., due to polysubstance use); cannabis consumption was confusingly described (e.g., cannabis-based treatments were indicated without specifying the actual nature of treatment); the symptoms of depression could not be differentiated from those of other disorders (e.g., anxiety and depression were studied by a common score); the study contained only data on general mood, wellbeing, internalizing symptoms, etc.; changes in depression were judged by participants only (e.g., by answering simple questions like “Did your depression improve?”). Importantly, we selected only those studies that investigated herbal cannabis, e.g., natural preparations that are commercially available, and for which the claims described in the first section were made. Studies employing purified or synthesized Δ9-tetrahydrocannabinol (THC), cannabidiol (CBD) or their mixture were excluded, similar to those on synthetic cannabinoids or on natural cannabis enriched by any means.

Our selection criteria are explained by the purpose of this review. We did not intend to investigate the role of the endocannabinoid system in depression, although we will briefly address this issue in the fourth chapter. The primary goal was to verify the justifiability of the hopes raised by some scientific publications and the media in people suffering from depression. From this point of view, the effects of synthetic or purified cannabinoids are irrelevant. On the one hand, synthetic/purified preparations are still illegal in most countries, and on the other, the media and patients are much less interested in their putative antidepressant effects than in the effects of natural cannabis. Studies on polysubstance use and those reporting mixed scores (e.g., depression + anxiety) were discarded to maintain the focus of the review. Finally, answers to simple questions like “Did your depression improve” were not considered because participants were not necessarily aware of the nature and symptoms of depression due to a lack of adequate training.

We found 156 studies that were eligible in terms of the criteria presented above.

## 3. Research Findings

More than half (81) of the studies investigated the interaction between cannabis consumption and depression cross-sectionally; 44 studies investigated the longitudinal associations of enhanced cannabis consumption, 9 investigated the impact of cannabis withdrawal, and 22 studied the longitudinal effects of medical cannabis treatment on the evolution of depressive symptoms. Almost all studies were purely observational, which made the establishment of consumption characteristics and sample sizes difficult. As regards the former, we used the terms employed by the authors (e.g., “heavy use”) even if the term was not quantified in the study. Where consumption characteristics were not specified, or where these differed widely across participants, we used the term “unclear” (see below). Regarding sample sizes, these were quite often variable across longitudinal studies, and sometimes, it was difficult to establish the actual sample size that substantiated a particular finding. Therefore, we indicated sample size ranges instead of precise sample sizes.

### 3.1. Cross-Sectional Studies

In theory, cannabis consumption should be negatively correlated with depression, as the latter is widely believed to be ameliorated by the former. At the same time, positive associations could not be excluded either, as cannabis consumption is frequently initiated by those with depressive mood. It was found that 79.0% of cross-sectional studies found a positive correlation between cannabis consumption and depression: those consuming cannabis showed higher levels of depression than those who did not consume cannabis (Table 1). The positive correlation was found in community samples as well as in participants who had a variety of social and health conditions, and in all ages and genders. This finding is theoretically consistent with the self-medication hypothesis, if we assume that the reason for consumption—depression—was still present in the users, because not enough time had passed for the antidepressant effect of cannabis to take effect.

The impact of the amount of cannabis consumed is difficult to establish because most studies did not address this issue, or their participants were highly different in this respect, without consumption habits being differentially investigated (59% of all cross-sectional studies). In other studies, consumption was characterized by terms that were seldom explained, e.g., “habitual”, “harmful”, “heavy”, “non-disordered”, “occasional”, “positive drug testing” (without separating participants based on the results of the tests), “problematic”, or “regular”. The vague description of consumption characteristics is likely explained by the heterogeneity of the large samples (up to over a million), temporal fluctuations in the consumption habits of consumers, and the lack of consumption records, which is understandable in such naturalistic studies. More precise figures were given in 12% of all cross-sectional studies. For instance, cannabis consumption was described as “daily”, “weekly”, “3 days a week”, or “daily for more than 30 days”, etc. Yet in other studies, consumers were characterized as having cannabis use disorder (13%), which may not directly specify the dose taken, but is indicative of the cannabis-related state of participants. Cannabis was positively associated with depression in all these cases, suggesting that consumption habits and the amounts of cannabis ingested had little extra impact on depression symptoms.

Cannabis consumption was not related to depression in 14.8% of the studies and was associated with low depression in 6.2%. The particulars of participants in these studies were not different from those seen in the studies where the association was positive (Table 1). This holds true for the particulars of consumption.

Although it was not always clearly stated, it transpires from most studies that cannabis consumption was enduring for most participants by the time of data sampling. This suggests that if participants used cannabis to relieve their depression, this goal was achieved only according to 6.2% of the studies. The opposite happened in 79.0% of studies, even though in three such studies, the participants consumed medical cannabis (Table 1).

### 3.2. Longitudinal Studies

Longitudinal associations were studied in three contexts. One group of studies explored the impact of emerging depression symptoms on cannabis consumption; the second group studied the impact of increased cannabis consumption on depression, whereas the third studied the consequences of cannabis withdrawal.

An increase in depression symptomatology was invariably followed by an increase in cannabis consumption (Table 2A). No conflicting results were reported. These findings are in line with the self-medication hypothesis, i.e., they suggest that once faced with depressive symptoms, participants turned to cannabis, likely to alleviate such symptoms. This can be explained, at least in part, by the widespread belief that cannabis alleviates depression (see Section 1). Expectations may be supported by the transient euphoric effects of cannabis, but not necessarily by its long-term consequences as shown by the studies reviewed here. Again, the association did not depend on the particulars of participants, e.g., on their age, gender, or social/health background. The increase in cannabis consumption was detected month or years after the emergence of depressive symptoms.

The initiation of cannabis consumption or its increase were associated with the aggravation of depression symptoms in 81.1% of studies (Table 2B). It would be interesting to see separately the impacts of initiation and increased consumption, but most studies did not differentiate the two. Depression symptoms were aggravated on a timescale of weeks to decades, most studies being performed several years after the change in cannabis consumption. Changes in consumption were usually evaluated by self-reports, but psychometric instruments were also employed. In a subgroup of studies (18.9%), changes in cannabis consumption did not show long-term associations with changes in depression, but importantly, no study showed a negative association. In other terms, no study reported that depression was alleviated after the initiation of cannabis use or after an increase in its consumption.

Withdrawal from cannabis consumption was followed by an amelioration of depression in 55.6% of studies (Table 3). The opposite (aggravation of depression after withdrawal) was reported by just one study (4.5%), whereas depression did not change in the remaining studies. The particulars of participants did not explain the variation of findings. Importantly, the efficacy of withdrawal was checked by drug screening in more than half of the studies.

Taken together, these findings suggest that cannabis increases rather than decreases depression, as an increase in cannabis consumption increased, whereas withdrawal from cannabis decreased depression in most studies. Unfortunately, there are no mechanistic studies addressing the latter phenomenon. In simple terms, one can hypothesize that the long-term use of cannabis increases depression, whereas the elimination of this depressogenic factor by withdrawal improves mood. More detailed/adequate explanations may be provided by further studies. It is important to note, however, that the complex pharmacology of natural cannabis makes such studies rather difficult (see Section 4.1).

### 3.3. Treatment with Medical Cannabis

We found 22 studies where cannabis was used to treat various conditions and where depressive symptoms were also evaluated (Table 4A–C). In such studies, the agents administered were called “medical cannabis”. It is important to note, however, that the term refers to the purpose of use rather than to a particular composition of herbal cannabis [152]. Cannabis preparations termed “medical” contain widely different amounts of the main constituents. For instance, the THC content of medical cannabis preparations of licensed Canadian producers varied from 0.14% to over 25% [153], and the CBD content of medical cannabis was similarly variable [154]. In addition, the term may refer to any product (dried flower, resin, tincture, capsule, etc.) or delivery system (inhalation, oral, sublingual, topical, etc.) [155]. The same was true for the preparations used in the studies reviewed here. The THC and CBD doses of medical cannabis intakes varied between 1–50 mg and 0–20 mg, respectively, whereas the THC:CBD ratio varied from overwhelmingly THC-dominant to overwhelmingly CBD-dominant within the same study [156,157]. The only exception was the study by Gambino et al. [158] where the dose varied by a factor of 4 only (10–40 drops of oil), and each participant received a preparation that contained 63 mg THC and 80 mg CBD per 10 mL oil. In most studies, however, participants consumed preparations of their own choice, and consequently, these were of varying composition. Dosage and the route of administration was also decided by the participants. Moreover, treatment details were not even given in many studies. In addition to herbal cannabis, participants received other treatments, which were explained by their medical condition. These included antidepressants in seven studies where herbal cannabis was effective. Noteworthy, the efficacy of medical cannabis vs. the combined treatment was not investigated. Treatments other than medical cannabis were not reported in eight of the studies where cannabis was effective.

The studies presented here, therefore, did not meet the classical requirements of clinical studies. Despite this deficiency, it is still remarkable that there was only one study where medical cannabis increased depression, there were only four where it did not affect it, while depression was ameliorated by the treatment in 77.3% of studies (Table 4A–C). This beneficial effect was reported in patients with a variety of conditions, including major depression. Importantly, we also calculated effect sizes where the published data made this possible, and these were rather large in some studies. Note that we used Hedges’ g for evaluating effect sizes because sample sizes were small in some studies, and this measure is preferable over Cohen’s d in such cases [159].

**Table 4 pharmaceuticals-17-00689-t004:** Treatment with medical cannabis.

A. Medical Marijuana Aggravated Depression								
Follow-Up	Instrument	Change	Condition	Effect Size	Age Class	N	Comparison Group	Ref.
5–6 years	Pharm	diagnostic	Chronic pain	-	18+	100+	no disorder	[160]
**B. Medical Marijuana Did Not Affect Depression**								
**Follow-Up**	**Instrument**	**Change**	**Condition**	**Effect Size**	**Age Class**	**N**	**Comparison Group**	**Ref.**
6 month	ESAS	nil	Various **	-	Ma/Eld	100+	none	[161]
12 weeks	HADS-D	nil	Various ***	-	18+	100+	no treatment	[162]
12 weeks	HADS-D	nil	Various ***	-	18+	100+	no treatment	[163]
variable	PHQ4	nil	Chronic pain	-	25+	100+	none †	[164]
**C. Medical Marijuana Alleviated Depression**								
**Follow-Up**	**Instrument**	**In→fin (dif.)**	**Condition**	**Effect Size**	**Age Class**	**N**	**Comparison Group**	**Ref.**
18 weeks	custom	6.9→3.8 (3.1)	Major depression	n.c.	Ya/Ma	<100	none	[165]
12 month	BDI	3.2→2.2 (1.0)	Chronic pain	0.59	19+	500+	none	[166]
various	BDI	18.0→11.0 (7.0)	Chronic pain	n.c.	Ma	500+	none	[167]
1 year	BDI	11.3→5.8 (5.5)	Community sample	0.79	Ma	<100	none	[168]
6 month	BDI	12.6→5.5 (6.9)	Chronic pain	0.87	18+	<100	none	[169]
3 month	DASS-21	15.2→10.7 (4.5)	Various ***	0.45	18+	2000+	none	[156]
3 month	ESAS	3.2→2.3 (0.9)	Various **	n.c.	Ma/Eld	100+	none	[161]
6 month	GDS	6.4→5.0 (1.4)	Various **	0.35	Eld	100+	none	[170]
4 weeks	GDS	9.0→3.0 (6.0)	BMS	n.c.	Eld	<100	none	[158]
variable	HADS-D	11.7→8.6 (3.1)	Anxiety/depre.	n.c.	Ma	500+	no treatment	[100]
9 month	HADS-D	4.3→3.8 (0.5)	Various ***	0.13	18+	100+	no treatment	[171]
various	HADS-D	not reported *	Various **	n.c.	Adol/Ma	100+	no treatment	[101]
3 month	PHQ-8	8.5→5.7 (2.8)	Chronic pain	0.52	Ma/Eld	<100	none	[157]
6 month	PHQ-9	12.0→7.0 (5)	Depres.	n.c.	Ya/Ma.	100+	none	[172]
2 years	PHQ-9	13.7→7.2 (6.5)	Various ***	1.03	Ma/Eld	5000+	none	[173]
~3 month	diagnostic	diagnostic	Multiple Sclerosis	n.c.	Ma	100+	none	[174]
various	PROMIS-29	61.6→57.5 (4.1)	Anxiety/PTSD	0.41	Ma	100+	none	[175]

Legend. * only beta coefficients given; ** mostly chronic non-cancer pain; *** pain, insomnia, anxiety, depression; †, some participants reported no use at follow-up (“cannabis naïve”), but their number was insignificant compared to the treated group (~5% of the sample); Adol, adolescent; BDI, Beck Depression Inventory; BMS, Burning Mouth Syndrome; custom, single-item, 10-point rating scale (validated); DASS-21, Depression, Anxiety, Stress Scale -21; diagnostic, change in the prevalence of mood disorders; Effect size, Hedges’ g; Eld, elderly; ESAS, Edmonton Symptom Assessment System; GDS, Geriatric Depression Scale; HADS-D, Hospital Anxiety and Depression Scale, depression subscale; Irritable Bowel Syndrome; In→fin, initial and final scores; Ma, Middle-aged; n.c., not calculable from the data provided; no treatment, usually also cannabis consumers; where checked, cannabis use was smaller than in the treated group. Pharm, depression identified by medications received; PHQ, Patient Health Questionnaire; the number indicates the question number variant of the test; PROMIS-29, Patient-Reported Outcomes Measurement Information System-29; Ya, young adult.

Findings obtained with medical cannabis were in sharp contrast with the findings obtained with cannabis in general. In contrast to the depressogenic effects of cannabis as shown by cross-sectional and longitudinal studies, medical cannabis appeared to alleviate depression in most studies, on a timescale of weeks to month.

A closer inspection of the reports suggests, however, that this conclusion is largely unfounded. The first reason is that concurrent antidepressant medications were allowed in seven studies (see above). The most important reason for questioning the conclusions is that out of the 17 studies where medical cannabis was effective, 14 did not use controls at all, whereas the remaining 3 used no-treatment controls only. This raises the possibility that the results were partly or entirely due to placebo effect.

To investigate this possibility, we reviewed placebo-controlled clinical studies published in the same period, which used similar psychometric instruments (Table 5). The search terms and eligibility criteria employed for selected studies were like those used for the cannabis–depression interaction, with three differences. The search term “cannabis [title/abstract]” was replaced with “placebo [title/abstract]”, we added a search term that referred to the psychometric instrument (e.g., BDI [title/abstract]) and we inspected only those reports that were freely available at our institute. We looked for those psychometric instruments that were used at least twice in medical cannabis studies, i.e., where an average was calculable. We searched the studies backwards, beginning with the most recent ones, and continued the search till we found twice as many studies as those studying medical cannabis (separately for each instrument). As expected, placebo also ameliorated depression over time; its effect sizes were large, and moreover, larger than the effect sizes of medical cannabis.

### 3.4. Overall Assessment

Both non-medical (“general”) and medical cannabis were studied in community samples as well as in a variety of social and medical conditions. Depression was investigated by various means, but all the studies reviewed here employed validated methodologies. This diversity is essentially beneficial because it reduced the chance that any effects of cannabis remained hidden. At the same time, it should be noted that the quality of the reviewed studies was quite low, primarily regarding the description of cannabis consumption. With one exception [158], no study went deeper than globally observing the effects of highly variable consumption patterns. This applies not only to the composition of the consumed cannabis, but also to the route of administration, although these can lead to different pharmacodynamic and pharmacokinetic profiles, potentially influencing the effects on depression. In many studies, cannabis consumption patterns were not even mentioned, and where the authors addressed this issue, they simply listed the observed patterns without grouping results based on them. In addition, the authors often (but not exclusively) relied on self-reported consumption, which raises concerns regarding recall bias and social desirability bias. It should be emphasized, however, that this review examined the veracity of the claim that cannabis is an efficient method of self-medication in depression. The studies constitute “real-world evidence”, as some authors formulated in the title of their study [161]. As such, they seemed appropriate for the purposes of this review.

In cross-sectional studies, a cannabis consumption history was typically associated with high depression. In most longitudinal studies, increased cannabis consumption was followed by an increase in depression symptoms, whereas withdrawal from cannabis ameliorated depression (Figure 1). These findings suggests that cannabis increases rather than treats depression. Albeit medical cannabis administration ameliorated depression over time, none of the studies performed over the last three years was placebo-controlled. This makes it impossible to disentangle potential placebo effects from the actual pharmacological action of cannabis, casting doubt on the observed antidepressant effects.

In clinical studies published in the same period, placebo also ameliorated depression, and the average effect size of placebo was larger than the average effect size of medical cannabis. Overall, the greater effectiveness of placebo raises further questions on the antidepressant effects of medical cannabis. Naturally, the comparison with placebo effects from other studies cannot substitute for direct evidence from controlled trials within the context of cannabis use. However, there is a shortage of such studies.

There was an important difference between the effects of “general” and medical cannabis. Whereas the former appeared to induce or aggravate depression in most studies, no such effect was observed with medical cannabis. The discrepancy cannot be due to the doses administered, route of administration or the THC/CBD ratio, because “general” and medical cannabis studies were rather similar in these respects. The only detectable difference between the findings presented in Table 2 and Table 4 is the duration of treatment. “General” cannabis was typically consumed for years (up to 20 years) in longitudinal studies, whereas the effects of medical cannabis treatment were typically assessed after a few weeks or months only. One can hypothesize that medical cannabis increased depression if the duration of treatment was sufficiently long. In support of this assumption, medical cannabis aggravated depression in one study, where effects were studied after 5–6 years [160]. Alternatively, medical cannabis increased depression in comparison with placebo if this was employed. Stronger placebo effects in other studies somewhat support this assumption. A more positive approach would suggest that the discrepancy was due to other factors such as patient selection and treatment context. If true, the antidepressant effects of medical cannabis were genuine, and depended on factors that were not studied so far. Naturally, all these are speculations only. At present, “general” cannabis seems to aggravate depression. Neither the antidepressant nor the depressogenic effect of medical cannabis is supported by the findings. More carefully performed placebo-controlled studies may clarify the issue.

## 4. Mechanistic Considerations

Neither previous reviews nor the current one found reliable evidence for the claim that cannabis relieves depression. Are there any theoretical considerations that would make the antidepressant effect of cannabis plausible? Despite the abundance of research on the relationship between cannabis and depression, this question was rarely if ever asked. This is probably because cannabis contains many psychoactive compounds, each of which affects multiple mechanisms. Although we are still far from understanding this complexity, the possibilities can perhaps be assessed, even if roughly only.

### 4.1. Neurochemistry

It is widely believed that THC, the most important psychotropic component of cannabis, mimics the effects of neural signaling molecules called endocannabinoids, particularly anandamide, and 2-arachidonoylglycerol (2-AG) as well as of other lipid messengers called N-acylethanolamines [196,197,198]. We recently described endocannabinoid mechanisms in detail, together with a brief overview of the history of discoveries [199]. Briefly, the effects of cannabinoids are exerted through two G-protein coupled receptors, CB1 and CB2 [200,201], both being present in the central nervous system [202,203]. The receptors are located presynaptically and inhibit neuronal signaling according to the retrograde inhibition concept [204,205,206]. The essence of this is that neurotransmission elicits the postsynaptic release of endocannabinoids, which bind to the presynaptic cannabinoid receptor, and inhibit the release of the neurotransmitter that elicited the process. As such, the cannabinoid receptors work as molecular brakes that limit neurotransmission when this exceeds certain limits.

These “molecular brakes” are expressed on the presynaptic membranes of a variety of neurons, including GABAergic, glutamatergic, serotonergic, cholinergic, dopaminergic, opioidergic, noradrenergic, and cholecystokinin neurons [207,208,209,210,211]. As such, endocannabinoids can limit the release of eight different neurotransmitters.

In addition to CB1 and CB2, endocannabinoids postsynaptically activate the vanilloid receptor type 1 (TRPV1) and other transient receptor potential channels (TRPV1-4; TRPA1, TRPM8) [212,213]. They are also active at the G-protein coupled receptors GPR55 and GPR18 [214,215,216], as well at a series of intracellular pathways that control basal neurotransmitter release, and the interaction between neurons and glia cells [217,218,219].

Importantly for this review, the affinity of THC to these binding sites is similar to, and sometimes higher than that of endocannabinoids [220,221,222,223], hence the assumption that these mechanisms mediate the effects of THC.

We are currently unable to directly link any molecular effects of THC to its potential antidepressant actions. However, the effects of THC are indirect, i.e., it acts by influencing other mechanisms. Therefore, it is possible to examine how the individual sub-mechanisms relate to the effects of known antidepressants or to the depression-related effects of specific ligands (Figure 2).

It occurs that there are no neurochemical effects of THC, which would be consistent with an antidepressant effect. For instance, the retrograde inhibition of glutamate release may be detrimental in depression as deficits in glutamatergic neurotransmission are considered to be the primary mediators of this psychiatric pathology [224]. Similarly, GABAergic deficits were associated with major depression; therefore, the retrograde inhibition of GABA release is more compatible with depressogenic rather than with antidepressant effects [225]. According to the serotonergic hypothesis of depression, diminished serotonergic neurotransmission should aggravate depression [226]. The role of noradrenergic neurotransmission is somewhat controversial as some effective antidepressants decrease brain norepinephrine levels [226]. Yet the selective noradrenaline reuptake inhibitor reboxetine, which promotes noradrenergic neurotransmission, is an effective antidepressant [227]. Diminished dopaminergic neurotransmission is believed to play a role in the development of major depression, whereas cholinergic dysfunctions may be responsible for the cognitive symptoms of depression [228,229]. Therefore, the reduction in dopamine or cholinergic neurotransmission may not be considered favorable in depressive states. Finally, TRPV1 antagonists were shown to have antidepressant properties, whereas cannabinoids act as agonists at these receptors [230].

The brief analysis provided above is naturally rudimentary for two reasons. Firstly, information on the neurochemical effects of THC and their interaction, dose dependence, brain distribution, etc., are poorly understood at present. Secondly, a more thorough discussion of the issue would stretch the scope of this review. Nevertheless, it is perhaps telling that the effects of THC on the neurochemical phenomena that play a major role in depression are the opposite of what we would expect from an antidepressant.

The effects of CBD are more favorable in terms of an antidepressant effect. For example, this active ingredient has positive effects on serotonergic signaling [231]. Based on this and the CBD literature, one would be inclined to assume that the effect of cannabis on depression is inversely proportional with the THC to CBD ratio. It can only be regretted that the experimenters paid so little attention to this issue, and even when addressed, findings were controversial. In one study, the antidepressant effect of cannabis depended on the presence of CBD in the preparation, whereas THC was considered responsible for the antidepressant effect in another [175,232]. Virtually all studies reported on the effects of THC-dominant preparations irrespective of the outcome of the study. As such, the relative roles of THC and CBD remain unclear.

It should also be noted that herbal cannabis has many active ingredients, not just THC and CBD. The terpene composition, for example, shows large differences between individual chemovars, and even with the same THC to CBD ratio, a differential terpene composition can lead to opposite effects on depression [233]. Unfortunately, however, very little is known about the depression-related effects of various terpenes.

One cannot rule out that the multitude of mechanisms result in an antidepressant effect by interaction, but the succinct presentation of the main neurochemical effects do not provide a theoretical framework that would make the antidepressant effects of herbal cannabis plausible.

### 4.2. Pharmacology

One can hypothesize that preclinical pharmacological findings have influenced therapeutic expectations for cannabis. For example, it was shown relatively early that knocking out the CB1 receptor results in depression-like behavior [234], while inhibition of the FAAH enzyme reduces depression [235]. Since the former inhibits whereas the latter enhances endocannabinoid signaling, it can be assumed in principle that depression can be ameliorated with exogenous cannabinoids. These and similar findings are often referred to when there is a need to substantiate the antidepressant effects of herbal cannabis with preclinical arguments. However, there are significant differences in how gene disruption, FAAH enzyme inhibition, and receptor ligands affect cannabinoid signaling.

The main components of the endocannabinoid system are not only the ligands and receptors, but also the synthesizing and degrading enzymes. The reason is that these signaling molecules are synthesized on demand, and after completing their role are transported back into the cytoplasm where they are degraded [236]. The rate-limiting synthesizing enzymes are N-acyl phosphatidylethanolamine-specific phospholipase D and diacylglycerol lipase for anandamide and 2-AG, respectively [237,238]. After accomplishing their role, endocannabinoids are taken up by a carrier-mediated transport. Within the cytoplasm, anandamide is degraded via fatty-acid amide hydrolase (FAAH), whereas 2-AG is degraded via monoacylglycerol lipase (MAGL) [239]. These processes are of great pharmacological importance.

Receptor ligands (e.g., THC) affect all those mechanisms that were listed above, and perhaps several others that were not mentioned above due to uncertainties regarding their nature. On its turn, CB1 gene disruption disrupts the flow of information through the CB1 receptor but does not affect mechanisms that are independent of these receptors. Such mechanisms appear to constitute about half of the mechanisms affected by receptor ligands (see Figure 2), which means that gene disruption is considerably more selective than receptor ligands. The most selective interventions, however, are those that target synthesizing or degrading enzymes. As endocannabinoids are synthesized on demand and are rapidly degraded after their release [236,240], they are present in the synapse for short periods only. The enzymes have no substrates between bouts of activation; therefore, their inhibition is inconsequential when the endocannabinoid system is inactive. This implies that enzyme, e.g., FAAH inhibitors augment physiological responses rather than induce non-physiological responses. The question that may be asked by the application of FAAH inhibition can be formulated as follows: “What happens if the natural endocannabinoid signaling was prolonged”? By contrast, receptor ligands answer a broader question: “What happens if all cannabinoid binding sites are affected at the level of the whole brain?”, whereas gene disruption allows answering the question “What happens if part of the mechanisms are eliminated?” [199].

Not surprisingly, there often are major differences in the effects of manipulations that belong to different classes. For instance, the disruption of the CB1 gene increased, whereas the pharmacologic blockade of the very same receptor by the antagonist SR141716A decreased anxiety; the blockade of the FAAH enzyme did not affect anxiety directly but altered the way in which challenges (e.g., stressors) were responded [241,242].

As such, not all the findings of pharmacological studies are supportive of, or are explanatory regarding the effects of herbal cannabis, not even those employing pure THC or CBD, because cannabis contains both and a series of other psychoactive agents in addition, which also influence the herb’s effect. Only those preclinical pharmacologic studies that use herbal cannabis per se make the antidepressant effects of cannabis plausible.

## 5. Conclusions

The hypothesis formulated in the first section appears false. The review of findings published over the last three years led to conclusions very similar to those that were based on studies performed in more unfortunate conditions, i.e., when cannabis was predominantly illegal, its production was mostly uncontrolled, and the idea of medical cannabis was very incipient only. It is worth stating that the questionable and even harmful effect of cannabis on depression is not necessarily relevant for the drug’s legal status. Its putative health effects are not limited to depression, and short-term or occasional use my not unavoidably worsen depression. However, it seems that the widely advertised antidepressant effect of cannabis is at variance with current scientific evidence, and those supporting this claim may mislead its current and prospective users.

Conclusions so unfavorable from the point of view of expectations were explained in various ways. Degenhardt et al. [243] suggested that the unfavorable association of cannabis consumption and depression may be due to common social, family, and contextual factors that increase risks of both heavy cannabis use and depression. The same authors also found relief in stating that even “If the relationship is causal, then on current patterns of cannabis use in the most developed societies cannabis use makes, at most, a modest contribution to the population prevalence of depression” (Degenhardt et al., [243], abstract). One may argue against these and similar claims. For instance, the enhancement of depression after the increase in cannabis consumption and the amelioration of depression after withdrawal are not supporting the “common cause theory”. One can also argue that the contribution of cannabis to the prevalence of depression in the “most developed societies” may depend on the prevalence of cannabis consumption. If the latter increased, its relative contribution to depression also increased. These, however, are only details. The essence of such defensive approaches is that their adherents contrast speculations with experimental results, enthusiasm with factual information. This is scientifically counterproductive. Explanations of unfavorable findings do not help treat depression.

In this respect, it is important to note that a non-negligible minority of studies are compatible with the antidepressant effects of cannabis. This may be linked to the fact that cannabis contains a wide variety of compounds, some of which have molecular effects compatible with antidepressant actions. It cannot be ruled out that products containing components in right proportions can indeed alleviate depression. Unfortunately, however, the authors paid little attention to the composition of the products, so the work to decipher the interaction between particular compositions and antidepressant effects has not even begun. On the other hand, the explanation my lie in the differential socioeconomic status, personality traits, genetic predispositions, environmental factors and co-occurring mental health conditions of study participants. These could significantly influence the relationship between cannabis and depression. Unfortunately, such factors have not been studied so far. Their careful analysis may be important for future research. Such research, however, has to deal with problems other than patient selection and the establishment of the effective cannabis chemovar and the proper route of administration. The prevalence of cannabis use disorder in people who use medicinal cannabis is comparable to that reported in people who use cannabis for recreational purposes [244]. Cannabis use disorder was frequent even with disorders where the efficacy of cannabis is more established than in the case of depression such as chronic pain, sleep, posttraumatic stress disorder and multiple sclerosis; moreover, rates were higher in some instances than in those who had no medical conditions [245,246,247,248]. Cannabis use disorder is a highly debilitating condition, and may per se limit the use of cannabis for treating depression.

An alternative for the quest for cannabis formulations with reliable antidepressant effects are targeted approaches that involve compounds with known effects on the endocannabinoid system. These may modulate the system in a more controlled and precise manner. Additionally, investigating the synergistic effects of cannabinoids and other compounds found in cannabis may provide new insights into the development of more personalized therapies. Hopes regarding such agents are not compromised by the inefficacy of herbal cannabis, e.g., by the disappointing effects of a very complex mixture of active agents, which, in addition, has a highly variable composition, and is currently administered in unestablished dose regimens and routes. The endocannabinoid system remains a major drug research target, including its involvement in depression. Yet herbal cannabis, as it regards the alleviation of depression, does not seem to live up to expectations.

## Figures and Tables

**Figure 1 pharmaceuticals-17-00689-f001:**
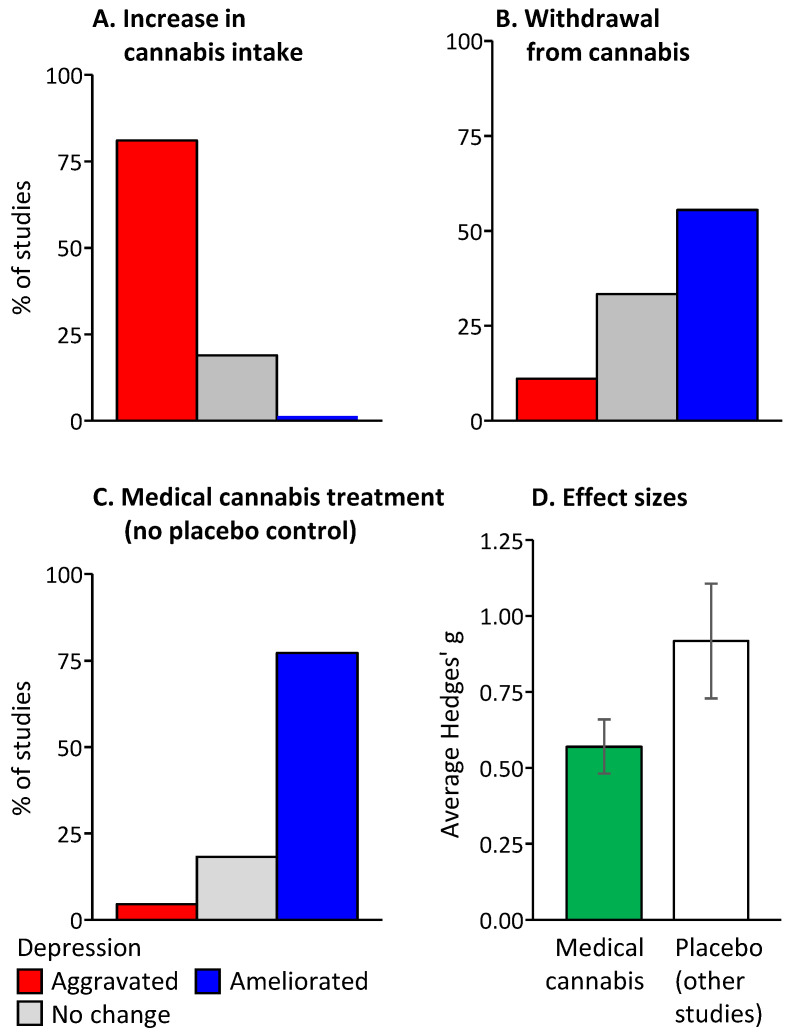
The main findings of the study.

**Figure 2 pharmaceuticals-17-00689-f002:**
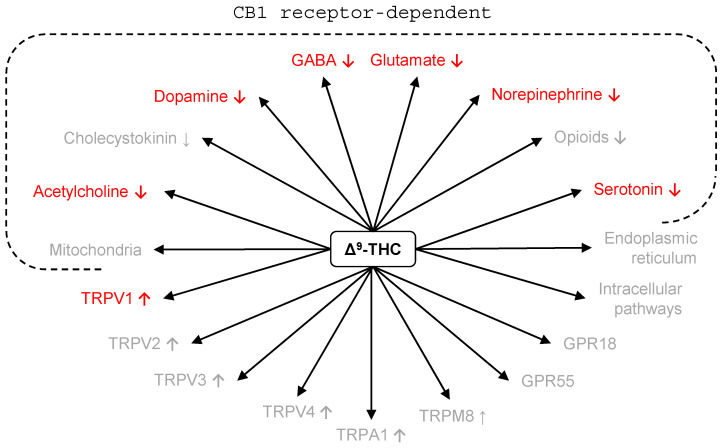
Neurochemical effects of THC and their relation to mechanisms underlying depression. ↓, decrease in function; red, the neurochemical effects of THC are inconsistent with antidepressant action; Gray, unknown or too complex to be discussed here (see text).

**Table 1 pharmaceuticals-17-00689-t001:** Cross-sectional associations between cannabis consumption and depression.

Cannabis Consumption Was Associated with High Depression					
Consumption Specifics	Sample Specifics	Gender at Birth	Age Class	Sample Size	Ref.
unclear	community sample	both	adolescent/young adult	100+	[26]
unclear	community sample	both	young adult	10,000+	[27]
unclear	community sample	both	adolescent	2000+	[28]
unclear	community sample	both	young adult	1000+	[29]
unclear	community sample	both	adolescent	1000+	[30]
unclear	community sample	both	18+	<100	[31]
unclear	community sample	both	adolescent	500+	[32]
unclear	community sample	men	adolescent/young adult	500+	[33]
unclear	community sample	both	adolescent	5000+	[34]
unclear	community sample	both	adolescent/young adult	2000+	[35]
unclear	community sample	both	18+	2000+	[36]
unclear	community sample	both	young adult/middle-aged	100+	[37]
unclear	community sample	both	young adult	500+	[38]
unclear	community sample	both	18+	10,000+	[39]
unclear	community sample	female	adolescent/young adult	2000+	[40]
unclear	community sample	both	16+	10,000+	[41]
lifetime use	community sample	both	adolescent	10,000+	[42]
lifetime use	community sample	both	adolescent	2000+	[43]
Non-disord-ered use	community sample	both	adolescent	10,000+	[44]
harmful	community sample	both	young adult	500+	[45]
>1/month for a year	community sample	both	adolescent/young adult	100+	[46]
>weakly	community sample	both	young adult	1000+	[47]
>weekly	community sample	both	adolescent/young adult	100+	[48]
daily for >30 days	community sample	both	18+	100,000+	[49]
Cannabis use disorder	community sample	both	adolescent	10,000+	[44]
unclear	Alzheimer disease	both	elderly	2000+	[50]
unclear	Army veterans	both	young adult	100+	[51]
unclear	Army veterans	both	young adult	1000+	[52]
Unclear medical	Army veterans	both	21+	2000+	[53]
>3 days/week	Army veterans	both	young adult	100+	[54]
cannabis use disorder	Army veterans	both	21+	2000+	[55]
unclear	Athletes	female	young adult	<100	[56]
cannabis use disorder	Bipolar disorder	both	18+	100,000+	[57]
unclear	Cancer	both	18+	10,000+	[58]
daily	Cancer survivors	both	middle aged	1000+	[59]
daily	Cancer survivors	both	18+	10,000+	[60]
cannabis use disorder	Cannabis use disorder	both	18+	1,000,000+	[61]
cannabis use disorder	Cannabis use disorder	both	young adult	1000+	[62]
heavy	Childhood adversity	both	18+	10,000+	[63]
unclear	Concussion	both	adolescent	10,000+	[64]
unclear	COVID19	both	adolescent	2000+	[65]
unclear	COVID19	both	18+	10,000+	[66]
unclear	COVID19, army veterans	both	young adult/middle-aged	1000+	[67]
lifetime use	COVID19	both	18+	2000+	[68]
abuse	COVID19	both	adults	100+	[69]
daily	COVID19	both	young adult	2000+	[70]
cannabis use disorder	Homelessness	both	adolescent/young adult	100+	[71]
unclear medical	Hospitalized (cannabis clinic)	both	18+	100+	[72]
unclear	Hospitalized (integrated care)	both	elderly	500+	[73]
unclear	Hospitalized (stress cardiomyopathy)	both	18+	10,000+	[74]
problematic use	Hospitalized (chronic disease)	both	18+	100+	[75]
Cannabis use disorder *	Hospitalized	both	adolescent	100,000+	[76]
unclear	Infertility	female	18+	100+	[77]
unclear	Inflammatory Bowel Disease	both	18+	1000+	[78]
unclear	Insomnia	both	adolescent	100+	[79]
unclear	Pregnancy	female	young adult/middle-aged	1000+	[80]
positive drug testing	Pregnancy	female	young adult/middle-aged	500+	[81]
cannabis use disorder	Pregnancy	female	young adult/middle-aged	1,000,000+	[82]
unclear medical	Primary care patients	both	18+	1,000,000+	[83]
cannabis use disorder	Primary care patients	both	18+	1,000,000+	[83]
unclear	PTSD COVID19	both	18+	100+	[84]
heavy	PTSD	both	18+	2000+	[85]
cannabis use disorder	PTSD	both	18+	10,000+	[86]
unclear	Various (mostly chronic pain)	both	18+	1000+	[87]
**Cannabis Consumption and Depression Did Not Associate**					
**Consumption specifics**	**Sample Specifics**	**Gender at Birth**	**Age Class**	**Sample Size**	**Ref.**
unclear	community sample	both	adolescent	2000+	[88]
unclear	community sample	both	young adult	500+	[89]
regular	community sample	both	15+	100,000+	[90]
daily	community sample	both	18+	10,000+	[91]
unclear	Binge eating	both	18+	100+	[92]
unclear	Cancer survivors	both	elderly	2000+	[93]
unclear	Hospitalized (bariatric surg.)	both	18+	500+	[94]
unclear	Juvenile offenders	males	adolescent/young adult	100+	[95]
unclear	Late chronotype	both	young adult	100+	[96]
daily medical	PTSD army veterans	both	middle-aged	100+	[97]
Cannabis use disorder *	Schizophrenia spectrum dis	both	18+	2000+	[98]
unclear	Substance use disorder	both	middle-aged	100+	[99]
**Cannabis Consumption Was Associated with Low Depression**					
**Consumption Specifics**	**Sample Specifics**	**Gender at Birth**	**Age Class**	**Sample Size**	**Ref.**
unclear	community sample	both	middle-aged	500+	[100]
occasional	community sample	both	18+	2000+	[36]
habitual	community sample	both	18+	2000+	[36]
unclear	IBD COVID19	both	middle-aged	500+	[101]
unclear	Psychosis first episode	both	child/adolescent	100+	[102]

Legend. *, alternatively, participants used cannabis more than twice a week; age classes, children (<11 years); adolescents (12–18); young adults (19–30); middle-aged (31–50); elderly (51+); No.+, age classes covered above the age shown by the number; COVID19, study performed during the COVID-19 pandemic; IBD, Inflammatory bowel disease; PTSD, posttraumatic stress disorder; sample size classes, n+, larger than n; <100, smaller than n; unclear, participants used and administered cannabis in various doses and routes, respectively, or consumption specifics were not detailed.

**Table 2 pharmaceuticals-17-00689-t002:** Longitudinal relationships between cannabis use and depression.

A. Cannabis Use Emerges or Increases after Depression					
Period Covered	Consumption Change	Sample Specifics	Age Class	Sample Size	Ref.
1 year	Self-reported	community sample	adolescents/young adults	500+	[103]
2 years	Self-reported	community sample	adolescents	1000+	[104]
Month *	Self-reported	COVID19	18+	100+	[105]
1 year	Self-reported	COVID19	adolescents/young adults	2000+	[106]
2 years	Self-reported	COVID19	middle-aged/elderly	2000+	[107]
1 year	Self-reported	Depression, major	elderly	10,000+	[108]
4 years	Diagnostic #	Army veterans	18+	1,000,000+	[109]
**B. Depression Emerges or Worsens after Increased Cannabis Use**					
**Period Covered**	**Consumption Change**	**Sample Specifics**	**Age Class**	**Sample Size**	**Ref.**
30 days	Self-reported	community sample	adolescents	1000+	[110]
1 year	Self-reported	community sample	adolescents/young adults	500+	[103]
1 year	Self-reported	community sample	adolescents	2000+	[111]
1 year	CDDUR	community sample	adolescents/young adults	<100	[112]
1 year	Self-reported	community sample	adolescents/young adults	2000+	[113]
2 years	Self-reported	community sample	adolescents/young adults	1000+	[114]
years-decades	Self-reported	community sample	adolescents	100,000+	[115]
4 years	ASSIST	‡ community sample	children	500+	[116]
4.5 years	Self-reported	community sample	young adults	2000+	[117]
~5 years	Self-reported	community sample	adolescents/young adults	100+	[118]
5 years	ASSIST	† community sample	young adults	1000+	[119]
5 years	CUDITr	♂ community sample	young adults	5000+	[120]
6 years	Self-reported	community sample	young adults	2000+	[121]
6 years	Self-reported	† community sample	young adults	2000+	[122]
12 years	Self-reported	♂ community sample	adolescents	1000+	[123]
~17 years	Self-reported	community sample	adolescents	5000+	[124]
20 years	Self-reported	community sample	adolescents	1000+	[125]
< 1 year	Self-reported	† Bipolar disorder	middle-aged	1000+	[126]
5 years	Diagnostic #	† Bipolar disorder	middle-aged	100+	[127]
1 year	Diagnostic #	Cannabis use disorder	18+	10,000+	[128]
1 year	Self-reported	COVID19	young adults	2000+	[129]
1 year	Self-reported	COVID19	elderly	10,000+	[130]
3 & 6 month	Self-reported	Hospitalized, surgery	middle-aged/elderly	1000+	[131]
30 days & 1 year	Self-reported	Hospitalized, bariatric surg.	middle-aged	5000+	[132]
2 years	Self-reported	Hospitalized, orthopedic surg.	18+	1000+	[133]
Month *	Medical records	♀ Pregnant	18+	500+	[134]
Month *	Diagnostic #	♀ Pregnant	young adults/middle-aged	100+	[135]
≥1 years	Self-reported	♀ Pregnant	young adults/middle-aged	500+	[136]
≥5 years	Self-reported	♀ Pregnant	young adults	100+	[137]
12 weeks	PXTSU	PTSD	young adults	1000+	[138]
**C. No Temporal Relationships between Depression and Cannabis Use**					
**Period Covered**	**Consumption Change**	**Sample Specifics**	**Age Class**	**Sample Size**	**Ref.**
~5 years	Self-reported	♂ community sample	young adults	500+	[139]
<16 years *	Self-reported	† community sample	adolescents	2000+	[140]
2 years	Self-reported	community sample	young adults	2000+	[141]
long term use	Self-reported	community sample	elderly	<100	[142]
20 years	Self-reported	community sample	adolescents	1000+	[125]
month *	Self-reported	COVID19	18+	100+	[105]
≤24 month *	Self-reported	psychosis, first episode	young adults	100+	[143]

Legend. #, changes in the prevalence of cannabis use disorder; *, temporal distance between study points varied; †, interaction between cannabis use and bipolar disorder; ‡, interaction between cannabis use and subclinical hypomania; age classes, children (<11 years); adolescents (12–18); young adults (19–30); middle-aged (31–50); elderly (51+); No.+, age classes covered above the age shown by the number; ASSIST, Alcohol, Smoking and Substance Involvement Screening test; CDDUR, Customary Drinking and Drug Use Record; COVID19, study performed during the COVID-19 pandemic; CUDITr, Cannabis Use Disorders Identification Test-Revised; gender at birth, ♂, males; ♀, females, no label, both; PTSD, posttraumatic stress disorder; PXTSU, Phenotype and eXposures Toolkit Substance Use; sample size classes, n+, larger than n; <100, smaller than n; Self-reported, answers to questions like “During the past 12 months, have you used hashish or marihuana?”.

**Table 3 pharmaceuticals-17-00689-t003:** Depression-related consequences of cannabis withdrawal.

Cannabis Withdrawal Decreased Depression						
Withdrawal Length	Withdrawal Check	Sample Specifics	Age Class	Sample Size	Comparison Group	Ref.
28 days	Drug screen	Multiple sclerosis	middle-aged	<100	no withdr.	[144]
few month	Drug screen	Pregnancy	young a./middle-a.	500+	no withdr.	[81]
1 year	Self-reported	Community sample	adolescents/young a.	10,000+	no withdr.	[145]
1 month	Self-reported	Cannabis use disorder	18+	<100	none	[146]
~5.5 years *	Self-reported	Cannabis use disorder	14+	100+	none	[147]
**Cannabis Withdrawal Had No Effect on Depression**						
**Withdrawal Length**	**Withdrawal Check**	**Sample Specifics**	**Age Class**	**N**	**Comparison Group**	**Ref.**
3 weeks	Drug screen	Community sample	young a.	<100	non-users	[148]
4 weeks	Drug screen	Community sample	adolescents/young a.	100+	none	[149]
7 years	Diagnostic #	Substance abuse	adolescents	1000+	none	[150]
**Cannabis Withdrawal Increased Depression**						
**Withdrawal Length**	**Withdrawal Check**	**Sample Specifics**	**Age Class**	**N**	**Comparison Groups**	**Ref.**
14 weeks	Drug screen	Cannabis use disorder	adolescents/young a.	<100	none	[151]

Legend. #, changes in the prevalence of cannabis use disorder; *, temporal distance between study points varied; age classes, adolescents (12–18 years); young a. (19–30); middle-a. (31–50); age classes: No.+, covered above the age shown by the number; sample size classes, n+, larger than n; <100, smaller than n; Self-reported, answers to simple questions.

**Table 5 pharmaceuticals-17-00689-t005:** Placebo effects in similar studies, which were reviewed because those on medical cannabis were not placebo controlled.

D. Placebo Alleviated Depression in Controlled Studies							
Follow-Up	Instrument	Initial→Final Score (dif.)	Condition	Effect Size	Age Class	N	Ref.
4 weeks	BDI	37.3→29.8 (7.5)	Major depression	0.71	children/adolescents	227	[176]
63 days	BDI	14.0→7.4 (6.6)	Constipation depression	1.00	middle-aged	60	[177]
5 weeks	BDI	16.6→8.4 (8.2)	Bariatric surgery	1.19	middle-aged	38	[178]
3 month	BDI	6.0→2.0 (4.0)	Lung cancer surgery	n.c.	middle-aged/elderly	156	[179]
8 weeks	BDI	13.5→5.5 (8.0)	Unexplained fatigue	n.c.	middle-aged	80	[180]
1 year	BDI	17.7→8.1 (9.6)	Coronary heart disease	n.c.	elderly	128	[181]
9 weeks	BDI	24.5→22.3 (2.2)	Community sample	n.c.	young adults/middle-aged	70	[182]
6 weeks	BDI	27.0→14.9 (12.1)	Depressive symptoms	n.c.	adolescent/young adults	62	[183]
12 weeks	GDS	7.0→5.8 (1.2)	Parkinson	0.37	middle-aged/elderly	171	[184]
1 month	GDS	8.0→3.0 (5.0)	Cardiac surgery	0.47	elderly	83	[185]
6 weeks	GDS	4.4→3.7 (0.7)	COPD	0.55	elderly	60	[186]
8 weeks	GDS	10.5→8.4 (2.1)	Major depression	0.76	elderly	117	[187]
3 month	GDS	10.5→8.4 (2.1)	Community sample	3.47	elderly	80	[188]
12 month	HADS-D	3.2→2.3 (0.9)	Prostate cancer	0.32	middle-aged/elderly	130	[189]
8 weeks	HADS-D	11.0→7.2 (3.8)	Functional dyspepsia	0.79	not specified	30	[190]
30 days	HADS-D	14.6→10.5 (4.1)	Irritable Bowel Syndrome	1.33	middle-aged	42	[191]
1 year	HADS-D	7.0→4.2 (2.8)	Coronary heart disease	n.c.	elderly	128	[181]
6 month	PHQ-9	3.9→1.9 (2.0)	Migraine	0.39	middle-aged	807	[192]
6 weeks	PHQ-9	12.2→8.8 (3.4)	Depression	0.59	18+	653	[193]
7 days	PHQ-9	15.2→13.0 (2.2)	Bowel Resection	0.73	young adults/middle-aged	120	[194]
8 weeks	PHQ-9	11.2→6.3 (4.9)	Atypical depression	1.11	middle-aged	200	[195]

Legend. age classes, children (<11 years); adolescents (12–18); young adults (19–30); middle-aged (31–50); elderly (51+); No.+, age classes covered above the age shown by the number; COPD, chronic obstructive pulmonary disease; n.c., effect size not calculable based on the data provided.

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
