# Peer review of "Herbal Cannabis and Depression: A Review of Findings Published over the Last Three Years"

_pharmaceuticals, 2024, doi:10.3390/ph17060689_

Round 1
Reviewer 1 Report
Comments and Suggestions for Authors
The effect of cannabis on depression appears to be complex and multifaceted, as indicated by the various studies included in the manuscript. The article provides a comprehensive review of the literature on the relationship between cannabis consumption and depression, focusing on both cross-sectional and longitudinal studies, as well as treatment with medical cannabis. While cross-sectional studies generally show a positive correlation between cannabis use and depression, suggesting that cannabis users tend to have higher depression levels, longitudinal studies reveal a more intricate picture. These studies indicate that an increase in cannabis consumption is often followed by a worsening of depression symptoms, while withdrawal from cannabis may lead to an improvement. Additionally, studies on medical cannabis treatment show mixed results, with some suggesting an amelioration of depression symptoms over time. However, the lack of placebo controls and variability in product composition raise questions about the validity of these findings. Overall, while there is some evidence supporting the antidepressant effects of medical cannabis, the broader consensus suggests that cannabis use is more likely to exacerbate depression symptoms rather than alleviate them.
The article is well-organized, with distinct sections for each type of study and clear subheadings within each section. The limitations in the methodologies of some studies are appropriately addressed, such as the lack of analytical data, variability in sample sizes, and vague descriptions of cannabis consumption characteristics.
Here are some comments for the author:
-The inclusion and exclusion criteria are well-defined. However, it would be helpful to provide a rationale/explanation for each criterion. For example, explain why studies on synthetic cannabinoids or those using purified or synthesized THC, CBD, or their mixtures were excluded.
-It's clear from the findings that an increase in depression symptoms is consistently followed by increased cannabis consumption, supporting the self-medication hypothesis. However, it would be helpful to discuss the potential mechanisms underlying this association. How might the pharmacological effects of cannabis interact with the symptoms of depression?
-The observation that withdrawal from cannabis consumption was followed by an improvement in depression in the majority of studies is noteworthy. It would be beneficial to explore the potential mechanisms underlying this phenomenon. Could it be related to the alleviation of withdrawal symptoms, or does it suggest a direct pharmacological effect on mood regulation?
-Considering the consistent findings across longitudinal studies, it's important to discuss the implications for clinical practice and public health policies. How could these findings influence the treatment of individuals with depression who use cannabis?
-The explanation of the term "medical cannabis" and the variability in its composition is informative. However, it would be supportive to elaborate on how this variability might impact the observed effects on depression. How do different THC:CBD ratios or delivery methods influence therapeutic outcomes?
-It's a significant result that some studies reported an amelioration of depression with medical cannabis. The contrast between the findings on medical cannabis and those on general cannabis use is striking. It is crucial to explore more possible explanations for this disparity. Discuss how factors such as patient selection, treatment context, or dosage might contribute to the observed differences.
-The examination of CBD's effects on depression-related neurochemistry adds nuance to the discussion. However, it would be helpful to elaborate further on how the presence of THC in cannabis products may counteract the potential antidepressant effects of CBD.
-The author could include in the Conclusion that researchers might explore more targeted approaches that involve isolating and utilizing specific compounds or formulations with known effects on the endocannabinoid system, instead of relying only on whole cannabis plant extracts. This could include developing drugs that mimic or modulate the endocannabinoid system's activity in a more controlled and precise manner. Additionally, investigating the synergistic effects of cannabinoids and other compounds found in cannabis may offer new insights into developing more tailored therapies.
Comments on the Quality of English Languageminor editing
Author Response
Reply to Reviewer 1
I thank the reviewer for his/her thorough and professional comments. Virtually all were observed.
Please find below our comments to the issues raised together with a clear indication of the changes made. In the manuscript, changes were printed in blue to easy their finding.
Comment 1. The inclusion and exclusion criteria are well-defined. However, it would be helpful to provide a rationale/explanation for each criterion. For example, explain why studies on synthetic cannabinoids or those using purified or synthesized THC, CBD, or their mixtures were excluded.
Reply. The reviewer is right; some of the criteria may not be self-evident. The issue was addressed in the before-the-last paragraph of section 2 (lines 104-115).
Comment 2. It's clear from the findings that an increase in depression symptoms is consistently followed by increased cannabis consumption, supporting the self-medication hypothesis. However, it would be helpful to discuss the potential mechanisms underlying this association. How might the pharmacological effects of cannabis interact with the symptoms of depression?
Reply. We believe that the main reason of self-medication is the widespread belief that cannabis relieves depression. The euphoria induced by cannabis seems to confirm this expectation. In other words, attempts at self-medication may have a partly social and partly a short-term pharmacological reason. Nevertheless, expectations are not necessarily supported by long-term effects as shown by the review. The issue was addressed I section 3.2. (lines 185-188).
Comment 3. The observation that withdrawal from cannabis consumption was followed by an improvement in depression in the majority of studies is noteworthy. It would be beneficial to explore the potential mechanisms underlying this phenomenon. Could it be related to the alleviation of withdrawal symptoms, or does it suggest a direct pharmacological effect on mood regulation?
Reply. This is an excellent remark. Unfortunately, we found no studies on the putative mechanisms of this consequence of cannabis withdrawal. At present stage, we must restrict ourselves at the simplest explanation possible: depression was ameliorated because a depressogenic factor – notably cannabis – was eliminated. Naturally, the issue may not be as simple as that, but there are no other explanations available at present. The issue was addressed in section 3.2. (lines 221-226).
Comment 4. Considering the consistent findings across longitudinal studies, it's important to discuss the implications for clinical practice and public health policies. How could these findings influence the treatment of individuals with depression who use cannabis?
Reply. I deliberately avoided this question, because I did not wish to load the manuscript with sociopolitical messages. I tried to stay on the level of facts throughout the review. However, the comment is legitimate, so I added a few comments to the review (see lines 501-506). Note that another reviewer also raised this issue. Hence the double labeling (blue and bold fonts).
Comment 5. The explanation of the term "medical cannabis" and the variability in its composition is informative. However, it would be supportive to elaborate on how this variability might impact the observed effects on depression. How do different THC:CBD ratios or delivery methods influence therapeutic outcomes?
Reply. See may reply to comment No. 7.
Comment 6. It's a significant result that some studies reported an amelioration of depression with medical cannabis. The contrast between the findings on medical cannabis and those on general cannabis use is striking. It is crucial to explore more possible explanations for this disparity. Discuss how factors such as patient selection, treatment context, or dosage might contribute to the observed differences.
Reply. Given the stronger effect of placebo in other studies one cannot state that medical cannabis decreased depression. It is true, however, that a depressant effect was not evident either. I discussed the issue in the new version (lines 344-363).
Comment 7. The examination of CBD's effects on depression-related neurochemistry adds nuance to the discussion. However, it would be helpful to elaborate further on how the presence of THC in cannabis products may counteract the potential antidepressant effects of CBD.
Reply. The available evidence does not support the assumption that THC counteracts the antidepressant effects of CBD. The effects of THC-dominant and CBD-dominant preparations were compared only as an exception, and even in these cases, contradictory findings were reported. The likely reason is that natural cannabis contains a series of other compounds, which affect depression on their own. These issues were addressed in lines 430-443.
Comment 8. The author could include in the Conclusion that researchers might explore more targeted approaches that involve isolating and utilizing specific compounds or formulations with known effects on the endocannabinoid system, instead of relying only on whole cannabis plant extracts. This could include developing drugs that mimic or modulate the endocannabinoid system's activity in a more controlled and precise manner. Additionally, investigating the synergistic effects of cannabinoids and other compounds found in cannabis may offer new insights into developing more tailored therapies.
Reply. Excellent suggestion. I used these thoughts in the last paragraph of the paper (lines 544-554).
Reviewer 2 Report
Comments and Suggestions for Authors
This manuscript presents a comprehensive review of recent literature investigating the relationship between herbal cannabis consumption and depression. While the topic is timely and relevant due to the increasing legalization and public interest in medical cannabis, several concerns require addressing before publication.
Main Question and Relevance:
The central question posed is whether recent changes in attitudes, legislation, and research practices have altered the understanding of cannabis's impact on depression compared to earlier findings. This is a crucial question as public perception often contrasts with scientific evidence, and the potential for cannabis as an antidepressant treatment holds significant public health implications. The review's focus on herbal cannabis specifically addresses a gap in the field often dominated by studies on isolated cannabinoids like THC and CBD.
Originality and Contribution:
The manuscript offers a valuable contribution by comprehensively analyzing recent studies on herbal cannabis and depression. This focus distinguishes it from earlier reviews that primarily addressed older research or concentrated on isolated cannabinoids. The inclusion of cross-sectional, longitudinal, and treatment studies provides a holistic perspective on the topic. Additionally, the manuscript delves into the pharmacological plausibility of cannabis's antidepressant effects, a dimension often neglected in similar reviews.
Methodology and Study Design:
While the review methodology is sound in its search strategy and inclusion/exclusion criteria, several limitations warrant attention:
· Heterogeneity of Studies: The included studies exhibit significant heterogeneity in sample characteristics, cannabis consumption patterns (often vaguely defined), and methodologies for assessing depression. This heterogeneity limits the ability to draw definitive conclusions and necessitates cautious interpretation of the findings. Any response on that?
· Lack of Placebo Controls: A critical concern is the absence of placebo controls in studies investigating the effects of medical cannabis on depression. This makes it impossible to disentangle potential placebo effects from the actual pharmacological action of cannabis, casting doubt on the observed antidepressant effects. Any take on that?
· Limited Consideration of Confounding Factors: The review acknowledges potential confounding factors, such as common risk factors for cannabis use and depression but does not adequately explore their impact on the observed associations. For example, socioeconomic status, personality traits, and co-occurring mental health conditions could significantly influence the relationship between cannabis and depression. Any elaboration on this?
Conclusions and Evidence:
The conclusions drawn, particularly regarding the depressogenic nature of herbal cannabis, are largely consistent with the evidence presented from cross-sectional and longitudinal studies. However, the claim of medical cannabis exhibiting antidepressant effects is not sufficiently supported due to the lack of placebo-controlled trials. The comparison with placebo effect sizes from other studies, while informative, cannot substitute for direct evidence from controlled trials within the context of cannabis use. Authors have to highlight this limitation.
Tables and Figures:
The tables comprehensively summarize the reviewed studies and are helpful for understanding the research landscape. Figure 1 effectively visualizes the main findings. However, Figure 2 requires further refinement to clarify the complex interplay between THC's neurochemical effects and depression-related mechanisms.
References:
The reference list is extensive and appears appropriate.
Caveats:
1. Overreliance on Self-Reported Data: The review predominantly focuses on studies relying on self-reported cannabis consumption. This raises concerns regarding recall bias and social desirability bias, where participants may underreport or misrepresent their cannabis use due to fear of judgment or legal repercussions. This is particularly relevant considering the historical context of cannabis prohibition and the ongoing stigma surrounding its use. Authors have to address this.
2. Lack of Consideration for Different Cannabis Products and Consumption Methods: The manuscript uses the term "herbal cannabis" broadly without adequately differentiating between various cannabis products (e.g., flowers, concentrates, edibles) and consumption methods (e.g., smoking, vaping, edibles). Each of these can lead to different pharmacodynamic and pharmacokinetic profiles, potentially influencing the effects on depression. A more nuanced analysis considering these variations is needed.
3. Limited Discussion of the Endocannabinoid System's Complexity: While the manuscript touches on the endocannabinoid system's role, it lacks a thorough discussion of its intricate involvement in mood regulation and depression. This includes the diverse functions of endocannabinoids beyond CB1 and CB2 receptors, such as their interaction with other neurotransmitter systems and their involvement in neuroplasticity and neuroinflammation.
4. Neglect of Potential Therapeutic Benefits for Specific Subgroups: The review focuses primarily on the overall relationship between cannabis and depression, potentially overlooking the possibility of therapeutic benefits for specific subgroups of individuals. For example, individuals with treatment-resistant depression or those with comorbid anxiety disorders might respond differently to cannabis. Further exploration of these nuances is necessary.
5. Lack of Discussion on Long-Term Effects and Potential Harms: The manuscript primarily focuses on short-term associations between cannabis and depression, neglecting the potential long-term effects of chronic cannabis use. This includes the risk of developing cannabis use disorder, cognitive impairments, and other mental health problems, which can exacerbate or contribute to depression.
6. Insufficient Attention to Individual Differences and Vulnerability Factors: The review does not adequately address the role of individual differences in vulnerability to the depressogenic effects of cannabis. Genetic predispositions, personality traits, and environmental factors can significantly influence how individuals respond to cannabis. This dimension requires further exploration.
Author Response
Reply to Reviewer 2
I thank the reviewer for his/her thorough and professional comments.
Please find below our comments to the issues raised together with a clear indication of the changes made. In the manuscript, changes were printed in blue to easy their finding.
Comment No 1. Heterogeneity of Studies: The included studies exhibit significant heterogeneity in sample characteristics, cannabis consumption patterns (often vaguely defined), and methodologies for assessing depression. This heterogeneity limits the ability to draw definitive conclusions and necessitates cautious interpretation of the findings. Any response on that?
Reply. I do not fully agree with this observation. Different authors performed the studies on different subjects, but within each study the sample was fairly homogeneous. Regarding the methods used to study depression, all studies used validated questionnaires. This diversity can be seen as an asset rather than a shortcoming. Only the incomplete description of cannabis consumption can be considered as a real deficiency. However, the review examines the veracity of the claims that cannabis reduces depression. These claims do not differentiate between different cannabis preparations, therefore the studies considered are suitable for verifying the truthfulness of the claim. These issues were discussed in the first paragraph of section “3.4. Overall assessment” (lines 310-322)
Comment No 2. Lack of Placebo Controls: A critical concern is the absence of placebo controls in studies investigating the effects of medical cannabis on depression. This makes it impossible to disentangle potential placebo effects from the actual pharmacological action of cannabis, casting doubt on the observed antidepressant effects. Any take on that?
Reply. I addressed this question several times in the manuscript, but this wording is more fortunate; therefore, I used it in lines 335-336 of the manuscript.
Comment No 3. Limited Consideration of Confounding Factors: The review acknowledges potential confounding factors, such as common risk factors for cannabis use and depression but does not adequately explore their impact on the observed associations. For example, socioeconomic status, personality traits, and co-occurring mental health conditions could significantly influence the relationship between cannabis and depression. Any elaboration on this?
Reply. Excellent point. I addressed this issue in lines 523-524 and 530-534 of the manuscript where I tried to explain why cannabis was efficient in a minority of studies.
Comment No 4. The conclusions drawn, particularly regarding the depressogenic nature of herbal cannabis, are largely consistent with the evidence presented from cross-sectional and longitudinal studies. However, the claim of medical cannabis exhibiting antidepressant effects is not sufficiently supported due to the lack of placebo-controlled trials. The comparison with placebo effect sizes from other studies, while informative, cannot substitute for direct evidence from controlled trials within the context of cannabis use. Authors have to highlight this limitation.
Reply. I highlighted the issue in lines 340-342 of section 3.4 Overall assessment.
Comment No 5. The tables comprehensively summarize the reviewed studies and are helpful for understanding the research landscape. Figure 1 effectively visualizes the main findings. However, Figure 2 requires further refinement to clarify the complex interplay between THC's neurochemical effects and depression-related mechanisms.
Reply. Fig. 2 was slightly changed to comply with the suggestions formulated in Comment No. 8.
Comment No 6. Overreliance on Self-Reported Data: The review predominantly focuses on studies relying on self-reported cannabis consumption. This raises concerns regarding recall bias and social desirability bias, where participants may underreport or misrepresent their cannabis use due to fear of judgment or legal repercussions. This is particularly relevant considering the historical context of cannabis prohibition and the ongoing stigma surrounding its use. Authors have to address this.
Reply. The reviewer is right. However, I must emphasize that (i) I did not select the studies in any way. I reviewed all studies that met the eligibility criteria. The reliance on self-reported cannabis use was not mine; it was chosen by the authors of the reviewed studies. (ii) The vast majority of studies were from states and countries where cannabis use was legal. As such, legal consequences are unlikely to affect the outcome of the studies. This and the following comment was addressed together in lines 322-328.
Comment No 7. Lack of Consideration for Different Cannabis Products and Consumption Methods: The manuscript uses the term "herbal cannabis" broadly without adequately differentiating between various cannabis products (e.g., flowers, concentrates, edibles) and consumption methods (e.g., smoking, vaping, edibles). Each of these can lead to different pharmacodynamic and pharmacokinetic profiles, potentially influencing the effects on depression. A more nuanced analysis considering these variations is needed.
Reply. Again, the reviewer is perfectly right. Unfortunately, however, my analysis could not go deeper than the studies reviewed. I found virtually no studies where the effects of particular preparations and administration routes were studied separately. These were not even mentioned in a good deal of the studies, and where the authors addressed this issue, they simply listed them without grouping results based on them. This and the previous comment was addressed together in lines 322-328.
Comment No 8. Limited Discussion of the Endocannabinoid System's Complexity: While the manuscript touches on the endocannabinoid system's role, it lacks a thorough discussion of its intricate involvement in mood regulation and depression. This includes the diverse functions of endocannabinoids beyond CB1 and CB2 receptors, such as their interaction with other neurotransmitter systems and their involvement in neuroplasticity and neuroinflammation.
Reply. See my reply to comment No. 5.
Comment No 9. Neglect of Potential Therapeutic Benefits for Specific Subgroups: The review focuses primarily on the overall relationship between cannabis and depression, potentially overlooking the possibility of therapeutic benefits for specific subgroups of individuals. For example, individuals with treatment-resistant depression or those with comorbid anxiety disorders might respond differently to cannabis. Further exploration of these nuances is necessary.
Reply. These issues were addressed in lines 501-506. The issue was raised by Reviewer 1 as well; hence the double highlight (bold and blue font).
Comment No 10. Lack of Discussion on Long-Term Effects and Potential Harms: The manuscript primarily focuses on short-term associations between cannabis and depression, neglecting the potential long-term effects of chronic cannabis use. This includes the risk of developing cannabis use disorder, cognitive impairments, and other mental health problems, which can exacerbate or contribute to depression.
Reply. The issue was addressed in lines 535-544. It was shown that the risk of cannabis use disorder is high with medical cannabis; moreover this risk is sometimes higher than with recreational cannabis use.
Comment No 11. Insufficient Attention to Individual Differences and Vulnerability Factors: The review does not adequately address the role of individual differences in vulnerability to the depressogenic effects of cannabis. Genetic predispositions, personality traits, and environmental factors can significantly influence how individuals respond to cannabis. This dimension requires further exploration.
Reply. See my replies to Comment 3. The reviewer is naturally right, but these issues were not studied so far; therefore, I cannot review them.
Reviewer 3 Report
Comments and Suggestions for Authors
Dear author, I have completed the review of your paper. Overall, it is good. I have some minor comments: Line 25: There is no need to write ‘the presentation of the problem’. Lines 32-33: Please rephrase the sentence as it is difficult to understand. Some abbreviations need to be written in full form, such as DSM and ICD. In Table 1, please correct the title to ‘CANNABIS CONSUMPTION AND DEPRESSION DID NOT ASSOCIATE’. In Table 2, the titles are not similar. Please correct this. In Table 2, please define what CDDUR stands for. In Table 3, specify the withdrawal effects of cannabis. Best regards
Comments on the Quality of English Languageplease rephrase some sentences in the text
Author Response
Dear Reviewer,
Many thanks for your detailed comments. Please find below our replies, together with a detailed account of the changes made to the manuscript. Changed text was printed in green to easy their finding.
Comment 1: Line 25: There is no need to write ‘the presentation of the problem’.
Reply. The title of the section was shortened to “1. A brief historical overview”.
Comment 2: Lines 32-33: Please rephrase the sentence as it is difficult to understand.
Reply. The sentence was rephrased together with the previous one (lines 29-33).
Comment 3: Some abbreviations need to be written in full form, such as DSM and ICD.
Reply. The abbreviations were replaced with full names (lines 88-89). Other abbreviations were also replaced with full names (see e.g. lines 462-463).
Comment 4: In Table 1, please correct the title to ‘CANNABIS CONSUMPTION AND DEPRESSION DID NOT ASSOCIATE’.
Reply. Done (see page 7, green text).
Comment 5: In Table 2, the titles are not similar. Please correct this.
Reply. Done (see page 9, green text)
Comment 6: In Table 2, please define what CDDUR stands for.
Reply. The abbreviation was explained in lines 207-208.
Comment 7: In Table 3, specify the withdrawal effects of cannabis.
Reply. The title of the table was changed to “Depression-related consequences of cannabis withdrawal” (line 227).
Round 2
Reviewer 1 Report
Comments and Suggestions for Authors
the authors covered all the issues I raised
Comments on the Quality of English Languageminor
Reviewer 2 Report
Comments and Suggestions for Authors
Glad with changes